# ReLiK: Retrieve, Read and LinK: Fast and Accurate Entity-Linking and Relation-Extraction on an Academic-Budget

## Abstract

Entity Linking (EL) and Relation Extraction (RE) are fundamental tasks in Natural Language Processing, serving as critical components in various applications such as Information Retrieval, Question Answering, and Knowledge Graph Construction. However, existing approaches often suffer from either a lack of flexibility, low-performance issues, or computational inefficiency. In this paper, we propose ReLiK, a Retriever-Reader architecture, where, given an input text, the Retriever module undertakes the identification of candidate entities or relations that could potentially appear within the text. Subsequently, the Reader module is tasked to discern the pertinent retrieved entities or relations and establish their alignment with the corresponding textual spans. Notably, we put forward an innovative input representation that incorporates the candidate entities or relations alongside the text, making it possible to link entities or extract relations in a single forward pass in contrast with previous Retriever-Reader-based methods, which necessitate a forward pass for each candidate. Our formulation of EL and RE achieves state-of-the-art performance in both in-domain and out-of-domain benchmarks while using academic budget training and with up to 40x inference speed with respect to other competitors. Finally, we propose a model for closed Information Extraction (cIE), i.e. EL + RE, which sets a new state of the art by employing a shared Reader that simultaneously extracts entities and relations.

## 1 Introduction

Extracting structured information from unstructured text lies at the core of many AI problems, such as Information Retrieval (Hasibi et al., 2016; Xiong et al., 2017), Knowledge Graph Construction (Clancy et al., 2019; Li et al., 2023), Knowledge Discovery (Trisedya et al., 2019), Automatic Text Summarization (Amplayo et al., 2018; Dong et al., 2022), Language Modeling (Yamada et al., 2020; Liu et al., 2020b), Automatic Text Reasoning (Ji et al., 2022), and Semantic Parsing (Bevilacqua et al., 2021; Bai et al., 2022), inter alia. Looking at the variety of applications in which IE systems are used, we argue such systems should strive to satisfy three fundamental properties: Speed, Flexibility, and Performance.

This work focuses on two of the most famous IE tasks: Entity Linking (EL) and Relation Extraction (RE). While tremendous progress has recently been made on both EL and RE, to the best of our knowledge, recent approaches only focus on at most two out of the three properties simultaneously, hindering their applicability in multiple scenarios. Here, we show that by harnessing the Retriever-Reader paradigm (Chen et al., 2017), it is possible to use the same underlying architecture to tackle both tasks, improving the current state of the art while satisfying all fundamental properties. Most importantly, our models are trainable on an academic budget with a short experiment lifecycle, leveling the current playing field and making research on these tasks accessible for academic groups.

We frame EL and RE similarly to recent Open-Domain question-answering (ODQA) systems (Zhang et al., 2023), where, given an input question, a bi-encoder architecture (Retriever) encodes the input text and retrieves the most relevant text passages from an external index containing their encodings. Then, a second encoder (Reader) takes in input the question and each retrieved passage separately and extracts the answer from a specific passage if present. Our framing differs from most

famous ODQA ones for two main reasons: i) for both EL and RE, the input text contains multiple questions simultaneously since there might be multiple entities to link, or multiple relations to extract; ii) we encode the input text with all its retrieved passages (i.e. the textual representations of the candidate entities or relations), linking all the entities or extracting all the relational triplets in a single forward pass. Our architecture can thus be conceptually divided into two main components:

- The **Retriever** that is tasked to retrieve the possible Entities/Relations that can be extracted from a given input text.

- The **Reader**, that, given the original input text and all the retrieved Entities/Relations (output of the Retriever), is tasked to connect them to the relevant spans in the text.

First, leveraging the non-parametric memory, i.e. the knowledge base, accessed by the Retriever component considerably lowers the number of parameters of the final model required to achieve state-of-the-art performances (Performance and Speed). Second, using textual representations for entities/relations combined with the Retriever component makes it easier for the final model to zero-shot on unseen entities/relations(Flexibility). Finally, leveraging the contextualization capabilities of novel large language models such as He et al. (2023), encoding the input text and the textual representation of entities/relation and linking/extracting them in the same forward pass improves both model's final performances and speed (Performance and Speed).

To foster research and usage ReLiK, we release the code at `http://www.omitted.link`.

## 2 BACKGROUND

**Entity Linking** (EL) is the task of identifying all the entity mentions in a given input text and linking them to an entry in a reference knowledge base. For example given the sentence "Michael Jordan was one of the best players in the NBA", a system performing EL and using the English Wikipedia as the reference knowledge base should be capable of linking the "Michael Jordan" span to the Wikipedia Page `en.wikipedia.org/wiki/Michael_Jordan` and the span containing "NBA" to `en.wikipedia.org/wiki/National_Basketball_Association`. Formally, we can define an EL system as a function that, given an input text $q$ and a reference knowledge base $\mathcal{E}$, identifies all the mentions along their corresponding entities $\{(m, e) : m \in \mathcal{M}(q), e \in \mathcal{E}\}$ where $m := (s, t) \in \mathcal{M}(q)$ represents a span within all the possible spans $\mathcal{M}(q)$ in the input text $q$ starting in $s$ and ending in $t$ with $1 \leq s \leq t \leq |q|$.

**Relation Extraction** (RE) is the task of extracting semantic relations between entities found within a given text from a closed set of relation types coming from a reference knowledge base. In the previous example sentence, a RE system using Wikidata as the reference knowledge base is expected to output triplets such as ("Michael Jordan", "NBA", `wikidata.org/wiki/Property:P118`), where P118 represents the relation type "league". Formally, for an input text $q$ and a closed set of relation types $\mathcal{R}$, RE consists of identifying all triplets $\{(m, m', r) : (m, m') \in \mathcal{M}(q) \times \mathcal{M}(q), r \in \mathcal{R}\}$ where $m$ and $m'$ are respectively the subject and object spans and $r$ a relation between them. The combination of both EL and RE as a unified task is known as closed Information Extraction (cIE).

## 3 THE READER-RETRIEVER (RR) PARADIGM

In this section, we introduce ReLiK our Retriever-Reader architecture for EL, RE, and cIE. While the Retriever is shared by all the tasks (Section 3.1), the Reader has a common formulation for span identification but slightly differs between the last linking and extraction steps (Section 3.2). Figure 1 shows a high-level overview of ReLiK as a unified framework for EL, RE and cIE.

### 3.1 RETRIEVER

We follow a retrieval paradigm similar to Dense Passage Retrieval (DPR) (Karpukhin et al., 2020) based on two encoders to produce a dense representation of our queries and passages. In our setup,

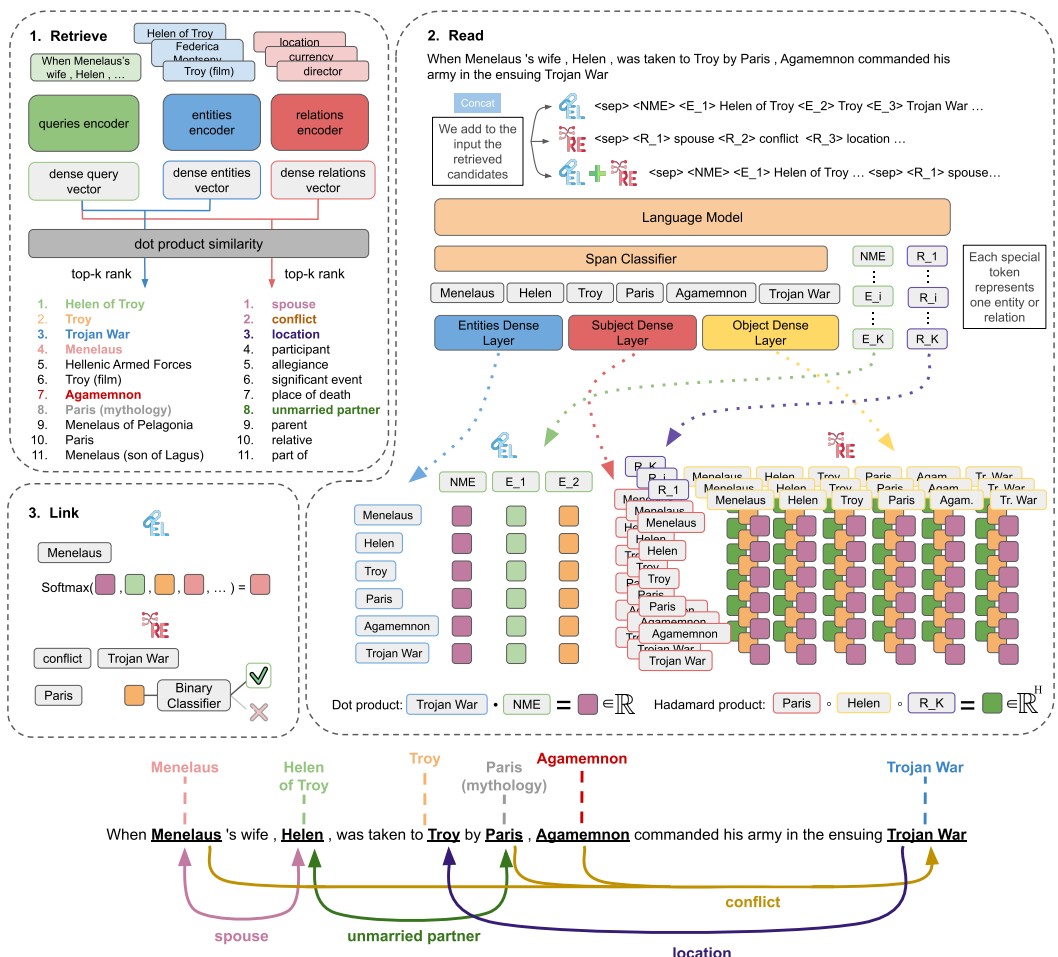

Figure 1: Description of ReLiK. Based on the RR-paradigm, we (1) Retrieve candidate entities and relations (2) Read and contextualize the text and candidates (3) Link and extract entities and triplets.

given an input text $q$ as our query and a passage $p \in \mathcal{D}_p$ in a collection of passages $\mathcal{D}_p$ that corresponds to the textual representations of either entities or relations, the Retriever model computes:

$$E_Q = \text{Retriever}_Q(\, q \,), \quad E_P = \text{Retriever}_P(\, p \,)$$

and ranks the most relevant entities or relations with respect to $q$ using the similarity function $\text{sim}(q, p) = E_Q(q)^\top E_P(p)$, where $\text{Retriever}_Q$ and $\text{Retriever}_P$ are Transformer encoders that compute the contextualized hidden representation of a query $q$ and a passage $p$ respectively.[1]

We train the Retriever employing a multi-label noise contrastive estimation (NCE) as training objective. The loss for $q$ is defined as:

$$\mathcal{L}_{Retriever} = -\log \sum_{p^+ \in \overline{\mathcal{D}_p}(q)} \frac{e^{\text{sim}(q, p^+)}}{e^{\text{sim}(q, p^+)} + \sum_{p^- \in P_q^-} e^{\text{sim}(q, p^-)}} \tag{1}$$

where $\overline{\mathcal{D}_p}(q)$ are the gold passages of the entities or relations present in $q$, and $P_q^-$ is the set of negative examples for $q$, constructed using gold passages from the other queries in the same minibatch and by hard negative mining using highest-scoring incorrect passages retrieved by the model

---

[1]The representations consist of the average of the encodings for the tokens in each of the two sequences.

## 3.2 READER

Differently from other ODQA approaches, our Reader performs a single forward pass for each input query. We append the top-k retrieved passages, $p_{1:K} = (p_1, \ldots, p_K), p_i \in \mathcal{D}_p$,[2] to the input query $q$, and obtain the following sequence, $q \, [SEP] \, \langle ST_0 \rangle \, \langle ST_1 \rangle \, p_1 \ldots \langle ST_K \rangle \, p_K$, with $[SEP]$ being a special token used to separate the query from the retrieved passages, and $\langle ST_i \rangle$ being special tokens used to mark the start of the $i$-th retrieved passage. We obtain the hidden representations $X$ of the sequence using a Transformer encoder:

$$X = \text{Transformer} \left( q \, [SEP] \, \langle ST_0 \rangle \, \langle ST_1 \rangle \, p_1 \ldots \langle ST_K \rangle \, p_K \right) \in \mathbb{R}^{l \times H} \tag{2}$$

where $l = |q| + 1 + (1 + K) + \sum_k |p_k|$ is the total length in tokens. Now, we predict all mentions within $q$, $\widetilde{\mathcal{M}}(q)$. We first compute the probability of the token $s$ to be the start of a mention as:

$$p_{start}(s|X) = \sigma_0(W_{start}^T X_s + b_{start}) \quad \forall s \in \{1 \ldots |q|\}$$

with $W_{start} \in \mathbb{R}^{H \times 2}, b_{start} \in \mathbb{R}^2$ being learnable parameters, and $\sigma_i$ the softmax function value at position $i$. Then the probability of a token $t$ to be the end of a mention with starting token $s$ is:

$$p_{end}(t|X, s) = \sigma_0(W_{end}^T X_m + b_{end}) \quad \forall t \in \{s \ldots |q|\}$$

with $W_{end} \in \mathbb{R}^{2H \times 2}, b_{end} \in \mathbb{R}^2$ being learnable parameters and $X_m \in \mathbb{R}^{2H}$ the concatenation of $X_s$ and $X_t$. We note that, with this formulation, we support the prediction of overlapping mentions. The loss for identifying mention spans in a single query is:

$$\mathcal{L}_{start} = - \sum_{s=0}^{|q|} \mathbb{1}_{\overline{\mathcal{M}_S}(q)}(s) log(p_{start}(s|X)) - \mathbb{1}_{\overline{\mathcal{M}_S}(q)^{\complement}}(s) log(1 - p_{start}(s|X))$$

$$\mathcal{L}_{end} = - \sum_{s \in \overline{\mathcal{M}_S}(q)} \sum_{t=s}^{|q|} \mathbb{1}_{\overline{\mathcal{M}}(q,s)}(t) log(p_{end}(t|X, s)) - \mathbb{1}_{\overline{\mathcal{M}}(q,s)^{\complement}}(t) log(1 - p_{end}(t|X, s))$$

Where $\overline{\mathcal{M}_S}(q)$ are the gold start tokens for the mentions in $q$ and $\overline{\mathcal{M}}(q, s)$ are the end tokens for mentions that start at $s$, $^{\complement}$ indicates complementary set and $\mathbb{1}$ is the indicator function. At inference time, we first compute all $s$ with $p_{start}(s|X) > 0.5$ and then all ends $p_{end}(t|X, s) > 0.5$ for each start $s$ to predict mentions $\widetilde{\mathcal{M}}(q)$.

While the formulation for extracting mentions from the input text is shared between EL and RE, the final steps to link them to entities and extract relational triplets are different. In what follows, we describe the two different procedures.

**Entity Linking** As we are now describing the EL step, in this paragraph the retrieved passages will identify the textual representations of the entities we have to link to the previously identified mentions, and thus we will change the notation of $p_{1:K} = (p_1, \ldots, p_K)$ to $e_{0:K} = (e_0, \ldots, e_K), e_{i \neq 0} \in \mathcal{E}$.[3] Specifically, for each $m \in \mathcal{M}(q)$, we need to find $\mathcal{E}(q, m)$, the entity linked to mention $m$. To do so, we use the hidden representations $X$ from Equation 2, and project each mention and special token in a shared dense space using a feed-forward layer:

$$M = \text{gelu} \left( W_{projection}^T X_m + b_{projection} \right)$$

$$E_{0:K} = \text{gelu} \left( W_{projection}^T [X_{\langle ST_{0:K} \rangle}, X_{\langle ST_{0:K} \rangle}]^T + b_{projection} \right)$$

Where $W_{projection} \in \mathbb{R}^{2H \times H}, b_{projection} \in \mathbb{R}^H$ are learnable parameters, and $[X_{\langle ST_{0:K} \rangle}, X_{\langle ST_{0:K} \rangle}] \in \mathbb{R}^{K \times 2H}$ represent the repetition along the hidden representation axis of the special tokens vectors $X_{\langle ST_{0:K} \rangle} \in \mathbb{R}^{K \times H}$ in order to match the shape of $X_m$. The probability of mention $m$ being linked to entity $e_k$ is computed as:

$$p_{ent}(\mathcal{E}(q, m) = e_k | M, E_{0:K}) = \sigma_k(E_{0:K}^T M) \quad \forall m \in \mathcal{M}(q), \, k \in \{0 \ldots K\}$$

---

[2]The k highest scoring passages according to the sim function introduced in Section 3.1

[3]Here $e_0$ symbolizes $NME$, i.e. mentions for which the gold entity is not in $\mathcal{E}$, represented by $\langle ST_0 \rangle$

Therefore, if $\overline{\mathcal{E}}(q, m)$ is the gold entity linked to $m$ in $q$, the loss for EL is:

$$\mathcal{L}_{EL} = -\log \sum_{m \in \overline{\mathcal{M}}(q)} \sum_{k=0}^{K} \mathbb{1}_{\overline{\mathcal{E}}(q,m)}(e_k) \log(p_{ent}(\mathcal{E}(q, m) = e_k | M, E_{0:K}))$$

To train ReLiK for EL, we optimize $\mathcal{L}_{EL}$ and the mention detection losses from 3.2: $\mathcal{L} = \mathcal{L}_{start} + \mathcal{L}_{end} + \mathcal{L}_{EL}$. At inference time we will have the predicted spans $\widetilde{\mathcal{M}}(q)$ as input to the EL module and we will take $\text{argmax}_k \, p_{ent}(\mathcal{E}(q, m) = e_k | M, E_{0:K})$ for each $m \in \widetilde{\mathcal{M}}(q)$ as its linked entity.

**Relation Extraction**    In this paragraph, the retrieved passages for an input text $q$ will instead identify the textual representations of relations $r_{1:K} = (r_1, \ldots, r_K), r_i \in \mathcal{R}$. Specifically for each pair of mentions, $(m, m') \in \mathcal{M}(q) \times \mathcal{M}(q)$ we need to find $\mathcal{R}(q, m, m')$, i.e. the relation types between $m$ and $m'$ in $q$. To do so, we use the hidden representations $X$ from Equation 2, and project each mention and special token using three feed-forward layers:

$$S_m = \text{gelu}\left(W_{subject}^T X_m + b_{subject}\right) \qquad O_{m'} = \text{gelu}\left(W_{object}^T X_{m'} + b_{object}\right)$$

$$R_k = \text{gelu}\left(W_r^T X_{\langle ST_k \rangle} + b_r\right)$$

Where $W_{subject}, W_{object} \in \mathbb{R}^{2H \times H}, W_r \in \mathbb{R}^{H \times H}, b_{subject}, b_{object}$ and $b_r \in \mathbb{R}^H$ are learnable parameters. We obtain a hidden representation for each possible triplet with the Hadamard product:

$$T_{m,m',k} = S_m \odot O_{m'} \odot R_k \in \mathbb{R}^H$$

Which is a dense representation of relation ($k$) between subject ($m$) and object ($m'$). Then, the probability that $m$ and $m'$ are in a relation $r_k$ in $q$ is:

$$p_{rel}(r_k \in \mathcal{R}(q, m, m') | T_{m,m',k}) = \sigma_0(W_{rel}T_{m,m',k} + b_{rel}) \,\forall\, (m, m') \in \mathcal{M}(q) \times \mathcal{M}(q), k \in \{1 \ldots K\}$$

With $W_{rel} \in \mathbb{R}^{H \times 2}, b_{rel} \in \mathbb{R}^2$ being learnable parameters. If we take $\overline{\mathcal{R}}(q, m, m')$ as the gold relations between $m$ and $m'$ in $q$, the loss for RE is defined as follows:

$$\mathcal{L}_{rel} = - \sum_{(m,m') \in \mathcal{M}(q) \times \mathcal{M}(q)} \left( \sum_{k=1}^{K} \mathbb{1}_{\overline{\mathcal{R}}(q,m,m')}(r_k) log(p_{rel}(r_k \in \mathcal{R}(q, m, m') | T_{m,m',k})) \right.$$

$$\left. - \mathbb{1}_{\overline{\mathcal{R}}(q,m,m')^\complement}(r_k) log(1 - p_{rel}(r_k \in \mathcal{R}(q, m, m') | T_{m,m',k})) \right)$$

To train ReLiK for RE we optimize $\mathcal{L}_{rel}$ and the mention detection from 3.2: $\mathcal{L} = \mathcal{L}_{start} + \mathcal{L}_{end} + \mathcal{L}_{rel}$. At inference time, we first compute all mentions $\widetilde{\mathcal{M}}(q)$, and then we predict all the triplets where $p_{rel}(r_k \in \mathcal{R}(q, m, m') | T_{m,m',k}) > 0.5 \,\forall\, (m, m') \in \widetilde{\mathcal{M}}(q) \times \widetilde{\mathcal{M}}(q)$.

**Closed Information Extraction**    In the previous paragraphs, we have described how to perform EL and RE separately with ReLiK. However, since both tasks share the same mention detection approach, ReLiK allows for closed IE with a single Reader. In this setup, we use the Retriever trained on each task separately to retrieve $e_{1:K} \in \mathcal{E}$ and $r_{1:K'} \in \mathcal{R}$. Then, the Reader performs both tasks at the same time. The only difference is the computation of the hidden representations in Equation 2 as:

$$X = \text{Reader}\left(q \, [SEP] \, \langle ST_0 \rangle \, \langle ST_1 \rangle \, e_1 \ldots \langle ST_K \rangle \, e_K [SEP] \, \langle ST_{K+1} \rangle \, r_1 \ldots \langle ST_{K+K'} \rangle \, r_{K'}\right)$$

Additionally, we leverage the predictions of the EL module to condition RE by taking

$$X_m = [X_s, X_t, \sigma(E_{0:K}^T M_m) X_{\langle ST_{0:K} \rangle}]$$

as the input to the RE module after EL predictions are computed. Notice that now $W_{subject}, W_{object} \in \mathbb{R}^{3H \times H}$. Finally, at training time the loss becomes $\mathcal{L} = \mathcal{L}_{start} + \mathcal{L}_{end} + \mathcal{L}_{el} + \mathcal{L}_{rel}$ for a dataset annotated with both tasks.

# 4    ENTITY LINKING

In this section, we describe the Experimental Setup (Section 4.1) and report on the results of our systems compared to current state-of-the-art solutions (Section 4.2) for EL.

## 4.1 EXPERIMENTAL SETUP

### 4.1.1 DATA

To evaluate ReLiK on Entity Linking, we reproduce the setting used by Zhang et al. (2022). We use the AIDA-CoNLL dataset (Hoffart et al., 2011, AIDA) for the *in-domain* training (AIDA train) and evaluation (AIDA testa for model selection and AIDA testb for test). The *out-of-domain* evaluation is carried on: MSNBC, Derczynski (Derczynski et al., 2015), KORE 50 (Hoffart et al., 2012), N3-Reuters-128, N3- RSS-500 (R500) (Röder et al., 2014), and OKE challenges 2015 and 2016 (Nuzzolese et al., 2015). As our reference knowledge base, we follow Zhang et al. (2022) and use the 2019 Wikipedia dump provided in the KILT benchmark (Petroni et al., 2021). We do not use any *mention-entities* dictionary to retrieve the list of possible entities to associate to a given mention.

### 4.1.2 COMPARISON SYSTEMS

We compare ReLiK with two autoregressive approaches, namely De Cao et al. (2021b), in which the authors train a sequence-to-sequence model to produce, given a text sequence in input, a formatted string containing the entities spans along with the reference Wikipedia title; and De Cao et al. (2021a) which build on top of this last approach by previously identifying the spans of text that may represent entities and then generate in parallel the Wikipedia title of each span, greatly enhancing the speed of the system. The most similar approach to our system is arguably Zhang et al. (2022), which was the first to invert the standard Mention Detection → Entity Disambiguation pipeline for EL. They first used a bi-encoder architecture to retrieve the entities that could appear in a text sequence and then an encoder architecture to reconduct each retrieved entity to a span in the text. We want to highlight that while the Retriever part of ReLiK for EL and Zhang et al. (2022) are conceptually the same, the Reader component strongly differs. Indeed, our Reader is capable of linking all the retrieved entities in a single forward pass, while theirs must perform a forward pass for each retrieved entity, being roughly 40 times slower to achieve the same performances. Finally, we note that, with the exception of Zhang et al. (2022), the other approaches use a *mention-entities* dictionary, i.e. a dictionary that for each mention contains a list of possible entities in the reference knowledge base to which the mention can be associated. In order to build such a dictionary for Wikipedia entities, the hyperlinks in Wikipedia pages are usually utilized Pershina et al. (2015). This means that given the input sentence "Jordan is an NBA player" in order to link the span "Jordan" to the Wikipedia page of Michael Jordan, there must be at least one page in Wikipedia in which a user manually linked that specific span (Jordan) to the Michael Jordan page. While for frequent entities, this might not represent a problem, for rare entities, it could mean the impossibility of linking them.

### 4.1.3 EVALUATION

We evaluate ReLiK on the GERBIL platform (Röder et al., 2018), using the implementation of Zhang et al. (2022) from the paper repository `https://github.com/WenzhengZhang/EntQA`. We report the results of evaluating the datasets described in Section 4.1.1 using the *InKB* F1 score with strong matching (predictions boundaries must match exactly gold ones).

### 4.1.4 ReLiK SETUP

**Retriever** We initialize the query encoder and passage encoder with $E5_{base}$ (Wang et al., 2022) pretrained on BLINK[4]. We train each encoder on the AIDA dataset for a maximum of 5000 steps using RAdam (Liu et al., 2020a) with a learning rate of 1e-5 and a linear learning rate decay schedule. We split each document into overlapping chunks of length $W = 32$ words with a stride $S = 16$, resulting in 12,995 windows in the training set, 3292 in the validation set, and 2950 in the test set. We concatenate to each window the first word of the document as in Zhang et al. (2022). We employ KILT (Petroni et al., 2021) to construct the entities index, which contains $|\mathcal{E}| = 5.9M$ entities. The textual representation of each entity is a combination of the Wikipedia title and opening text for the corresponding entity contained within KILT. We optimize the NCE loss (Equation 1) with 400 negatives per batch. At the end of each epoch, we mine at most 15 hard negatives per sample in the batch among the highest-scoring incorrect entities retrieved by the model. Appendix A.2.1 shows all the parameters used during the training process.

---

[4]Appendix A.1 provides details on the pretraining process.

| | **In-domain** | **Out-of-domain** | | | | | | | | **Avgs** | | |
|---|---|---|---|---|---|---|---|---|---|---|---|---|
| Model | AIDA | MSNBC | Der | K50 | R128 | R500 | O15 | O16 | Tot | OOD | AIT (m:s) |
| De Cao et al. (2021b)† | 83.7 | 73.7 | 54.1 | 60.7 | 46.7 | 40.3 | 56.1 | 50.0 | 58.2 | 55.0 | 38:00 |
| De Cao et al. (2021a)†* | 85.5 | 19.8 | 10.2 | 8.2 | 22.7 | 8.3 | 14.4 | 15.2 | — | — | 00:52 |
| Zhang et al. (2022) | 85.8 | 72.1 | 52.9 | 64.5 | **54.1** | 41.9 | 61.1 | 51.3 | 60.5 | 57.3 | 20:00 |
| ReLiK$_B$ | 85.9 | 71.9 | 55.5 | 67.2 | 49.2 | 41.5 | 62.6 | 53.9 | 61.0 | 57.9 | 00:29 |
| ReLiK$_L$ | **86.5** | **74.2** | **56.6** | **73.9** | 51.4 | **43.0** | **66.1** | **55.4** | **63.4** | **60.5** | 01:46 |

Table 1: Comparison systems' evaluation (*inKB Micro $F_1$*) on the *in-domain* AIDA test set and *out-of-domain* MSNBC (MSN), Derczynski (Der), KORE50 (K50), N3-Reuters-128 (R128), N3-RSS-500 (R500), OKE-15 (O15), and OKE-16 (O16) test sets. **Bold** indicates the best model and underline indicates the second best competitor. † mark systems that use mention dictionaries. For De Cao et al. (2021a), we report the results on the Out-of-domain benchmark running the model from the official repository, but without using any *mention-entity* dictionary since no implementation of it is provided. AIT column shows the time in minutes and seconds (m:s) that the systems need to process the whole AIDA test set using a NVIDIA RTX 4090, except for Zhang et al. (2022) that does not fit in 24GB of RAM and for which an A100 is used.

**Reader** We train the Reader model with the windows produced by the Retriever on the AIDA dataset. While in the Retriever we use the Wikipedia openings as the entities' textual representations, in the Reader, due to computational constraints, and as in other works (De Cao et al., 2021b;a), we use Wikipedia titles, which demonstrated to be informative and discriminative in most situations (Procopio et al., 2023). In order to handle the long sequences created by the concatenation of the top-100 retrieved candidates to the windows, we use DeBERTa-v3 (He et al., 2023) as our underlying encoder. We train two versions of it using DeBERTa-v3 base (183M parameters, ReLiK$_B$) and DeBERTa-v3 large (434M parameters, ReLiK$_L$). We optimize both ReLiK$_B$ and ReLiK$_L$ using AdamW and apply a learning rate decay on each layer as in Clark et al. (2020) for 50,000 optimization steps. A table with all the training hyperparameters can be found in Appendix A.2.1.

## 4.2 RESULTS

**Performance** We show in Table 1 the *InKB F1* score ReLiK and its alternatives attein on the evaluation datasets. Arguably, the most interesting finding we report is the improvement in performance we achieve with respect to Zhang et al. (2022). Indeed, not only even ReLiK$_B$ outperforms Zhang et al. (2022) both in- and out-of-domain (85.9 vs 85.8 in-domain and 57.9 vs 57.3 Avg. out-of-domain) with fewer parameters (289M parameters vs 650M parameters), but it does so using a single forward pass to link all the entities in a window of text, greatly enhancing the final speed of the system. A broader look at the table shows that ReLiK$_L$ surpasses all its competitors on all evaluation datasets except R128, thus setting a new state of the art. Finally, another interesting finding is ReLiK$_L$ outperforming its best competitor by 9.4 points on K50. While the other datasets contain news and encyclopedic corpora annotations, K50 is specifically designed to capture hard-to-disambiguate mentions that involve a deep understanding of the context in which they appear. A qualitative Error Analysis of the predictions can be found in Appendix A.5.

**Speed and Flexibility** As we can see from Table 1, ReLiK$_B$ is the fastest system among the competitors. Not only that, the second fastest system, De Cao et al. (2021a), requires a *mention-entities* dictionary that contains the possible entities to which a mention can be linked. When not using such a dictionary, the results on the AIDA test set drop by 43% (De Cao et al., 2021a) and, as reported in Table 1, it becomes unusable in out-of-domain settings. We want to stress that systems that leverage such dictionaries are less flexible in predicting unseen entities during training and, most importantly, cannot link at all entities to mentions to which they are not specifically paired in the reference dictionary. Finally, our formulation allows the use of relatively large language models such as DeBERTa-v3 large and achieves unprecedented performance while keeping competitive inference speed. Report and ablations on ReLiK efficiency can be found in Appendices A.3 A.4.

# 5 Relation Extraction and closed Information Extraction

In this section, we first present the Experimental Setup (Section 5.1) for RE and cIE, and compare the results of our systems to the current state of the art (Section 5.2).

## 5.1 Experimental Setup

### 5.1.1 Data

**RE**  We choose two of the most popular datasets available. NYT (Riedel et al., 2010), which has 24 relation types, 60K training sentences and 5K for validation and test; and CONLL04 (Roth & Yih, 2004) with 5 relation types, 922 training sentences, 231 for validation and 288 for testing.

**cIE**  We follow previous work and report on the REBEL dataset (Huguet Cabot & Navigli, 2021), which leverages entity labels from Wikipedia and relation types (10,936) from Wikidata. We sub-sample 3M sentences for training and 10K for validation with the same test set as Josifoski et al. (2022) containing 175K sentences.

### 5.1.2 Comparison Systems

**RE**  We compare ReLiK with recent state-of-the-art systems for RE. Initial RE models performed NER and RC separately, disregarding the interaction between the two. As with EL, a recent trend in RE has been seq2seq approaches. Huguet Cabot & Navigli (2021) reframed the task as a triplet sequence generation, in which the model learns to *translate* the input text into a sequence of triplets. Lu et al. (2022) followed a similar approach to tackle several IE tasks, including RE. They were the first to include labels as part of the input to aid generation. However, while these approaches were flexible and end-to-end, they suffer from efficiency, as they are autoregressive. Lou et al. (2023) built upon Lu et al. (2022), dropping the need for a Decoder by having labels as part of the input and reframing the task as linking mention spans and labels between each other, pairwise. Lou et al. (2023) is somewhat similar to our EL Reader component. However, their approach does not include a Retriever, limiting the number of relation types that can be predicted, and their linking pairwise strategy leads to ambiguous decoding for triplets (See A.6 for more details).

**cIE**  Closed Information Extraction has been traditionally tackled using pipelines with systems trained separately for EL and RE. Recently, Josifoski et al. (2022) presented an autoregressive approach inspired by Huguet Cabot & Navigli (2021) in which the triplets decoded contains the unique Wikipedia title of each entity instead of their surface form with the aid of constraint decoding as in De Cao et al. (2021b). Rossiello et al. (2023) extended their approach by outputting both surface forms and titles. As with RE, autoregressive approaches did lift the ceiling for cIE, however, they are still slow and computationally heavy at inference time.

### 5.1.3 Evaluation

We report on micro-F1, using boundaries evaluation, i.e. a triplet is considered correct when entity boundaries are properly identified along the relation type. For cIE, we consider a triplet correct only when both entity spans, their disambiguation, and the relation type between both are correct. To ensure a fair comparison with previous autoregressive systems, we only consider entities present in triplets for EL, albeit ReLiK is able to disambiguate all of them.

### 5.1.4 ReLiK Setup

**Retriever**  As in the EL setting (Section 4.1.4), we initialize the query and passage encoders with E5 (Wang et al., 2022). In this context, we utilize the `small` version of E5. This choice is driven by the limited search space in contrast to the Entity Linking setting. Consequently, this enables us to significantly lower the computational demands for both training and inference. We train each encoder for a maximum of 40,000 steps using RAdam (Liu et al., 2020a) with a learning rate of 1e-5 and a linear learning rate decay schedule. For NYT we have $|\mathcal{R}| = 24$ and for REBEL we use all Wikidata properties with their definitions, $|\mathcal{R}| = 10,936$, and for EL we use the same settings

| Model | Params. | NYT | | CONLL04 | | REBEL | |
|---|---|---|---|---|---|---|---|
| | | | Pretr. | | Pretr. | EL | RE |
| Huguet Cabot & Navigli (2021) | 460M | 93.1 | 93.4 | 71.2 | 75.4 | — | — |
| Lu et al. (2022) | 770M | 93.5 | — | 71.4 | 72.6 | — | — |
| Lou et al. (2023) | 355M | 94.0 | 94.1 | 75.9 | **78.8** | — | — |
| Josifoski et al. (2022) | 460M | — | — | — | — | 79.7 | 68.9 |
| Rossiello et al. (2023) | 460M | — | — | — | — | 82.7 | 70.7 |
| ReLiK$_S$ | 33M + 141M | 94.4 | 94.4 | 71.7 | 75.8 | 83.7 | 73.8 |
| ReLiK$_B$ | 33M + 183M | 94.8 | 94.7 | 72.9 | 77.2 | 84.1 | 74.3 |
| ReLiK$_L$ | 33M + 434M | **95.1** | 94.9 | 75.0 | 78.1 | **85.1** | **75.6** |

Table 2: Micro-F1 results for systems trained on NYT, CONLL04 and REBEL datasets. Params. column shows the number of parameters for each system. EL reports only on entities belonging to a triplet. Pretr. indicates the model underwent pretraining on additional task-specific data.

explained at Section 4.1 with KILT as KB, $|\mathcal{E}| = 5.9M$. We optimize the NCE loss (1) using 24 negatives per batch for NYT and 400 for REBEL. More details included in Appendix A.2.1.

**Reader** The Reader setup mirrors that of EL. We use the DeBERTa-v3 model in all three sizes with AdamW as the optimizer and a linear decay schedule. For NYT we set $K = 24$, effectively utilizing the Retriever as a ranker. For the CONLL4 dataset, we use the NYT's Retriever. We explore a setup where ReLiK is pretrained using data from REBEL and NYT[5]. In the context of closed Information Extraction (cIE) we set $K = 25$ and $K' = 20$ as the number of passages for EL and RE respectively. In all cases, we select the best-performing validation step for evaluation. A table with all the parameters utilized during training can be found in Appendix A.2.1.

## 5.2 RESULTS

**RE** In Table 2, we present the performance of ReLiK in comparison to other systems. Notably, on NYT ReLiK$_S$ achieves remarkable results, outperforming all previous systems while utilizing fewer parameters and remarkable speed, around 10 seconds to predict the entire NYT test set (see Appendix A.3 for more details). The only exception is the CONLL04 dataset, where ReLiK is outperformed by Lou et al. (2023). However, it's important to note that CONLL04 is an extremely small dataset, where a few instances lead to a big gap in performance.

**cIE** The right side of Table 2 reports on closed Information Extraction. Here, ReLiK truly shines as the first efficient end-to-end system for jointly performing EL and RE with exceptional performance. It not only outperforms previous approaches in all its model sizes by a significant margin but it is also up to 35x times faster (see Appendix A.3 for more details). ReLiK enables cIE in real-world downstream applications in a previously unattainable capacity.

A qualitative Error Analysis of the predictions can be found in Appendix A.5.

## 6 CONCLUSION

In this work, we presented ReLiK, a novel and unified Retriever-Reader architecture that seamlessly attains state-of-the-art performance for both Entity Linking and Relation Extraction. Furthermore, taking advantage of the common architecture and using a shared Reader, our system is capable of achieving unprecedented performance and efficiency even on the closed Information Extraction task (i.e. Entity Linking + Relation Extraction). Our models are considerably lighter, an order of magnitude faster, and trained on an academic budget. We believe that ReLiK can advance the field of Information Extraction in two directions: first, by providing a novel framework for unifying other IE tasks beyond EL and RE, and, second, by providing accurate information for downstream applications in an efficient way.

---

[5]We replicate the approach from USM by sampling 300K from REBEL dataset plus NYT train set. We pretrain for 250,000 steps with the same settings as NYT.

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

# A  APPENDIX

## A.1  RETRIEVER PRETRAIN FOR ENTITY LINKING

We pretrain the Retriever on BLINK (Wu et al., 2020) employing a single-encoder architecture in which we initialize a E5$_{base}$ encoder to act both as the query encoder and the passage encoder. We split each document $d$ in overlapping windows $q$ of $W = 32$ words with a stride $S = 16$. To reduce the computational requirements, we (1) random subsample 1 million windows from the entire BLINK dataset, and (2) we retrieve hard negatives at each 10% of an epoch.We employ the same strategy as in Section 4.1.4 to construct the entity index, namely, we utilize KILT (Petroni et al., 2021) as our knowledge base, and we construct the textual representation of each entity within KILT by concatenating the Wikipedia title and opening text. We optimize the NCE loss (1) with 400 negatives per batch. At each hard-negatives retrival step we mine 15 hard negatives per sample in the batch with a probability of 0.2 among the highest-scoring incorrect entities retrieved by the model. We train the encoder for a maximum of 110,000 steps using RAdam (Liu et al., 2020a) with a learning rate of 1e-5 and a linear learning rate decay schedule.

## A.2  EXPERIMENTAL SETUP

### A.2.1  HYPERPARAMETERS

**Retriever**  We report in Table 3 the hyperparameters we used to train our Retriever for both Entity Linking and Relation Extraction.

**Reader**  We report in Table 4 the hyperparameters we used to train our Reader for both Entity Linking and Relation Extraction.

### A.2.2  IMPLEMENTATION DETAILS

We implement our work in PyTorch (Paszke et al., 2019), using PyTorch Lightning (Falcon & The PyTorch Lightning team, 2019) as the underlying framework. We use the pretrained models for E5 and DeBERTa-v3 from HuggingFace Transformers (Wolf et al., 2020).

### A.2.3  HARDWARE

We train every model on a single NVIDIA RTX 4090 graphic card with 24GB of VRAM.

| Hyperparameter | Values | | |
|---|---|---|---|
| | BLINK | EL | RE |
| Optimizer | RAdam | RAdam | RAdam |
| Learning Rate | 1e-5 | 1e-5 | 1e-5 |
| Weight Decay | 0.01 | 0.01 | 0.01 |
| Training Steps | 110,000 | 5000 | 40,000 |
| Patience | 0 | 5 | 5 |
| Query Batch Size | 64 | 64 | 64 |
| Max Query Length | 64 | 64 | 64 |
| Passage Batch Size | 400 | 400 | [24, 400] |
| Max Passage Length | 64 | 64 | 64 |
| Hard-Negative Probability | 0.2 | 1.0 | 1.0 |

Table 3: Hyperparameter we used to train the Retriever for the Entity Linking Pretrain (BLINK), Entity Linking (EL), and Relation Extraction (RE).

| Hyperparameter | Values | | | |
|---|---|---|---|---|
| | AIDA | NYT | CONLL04 | REBEL |
| Optimizer | AdamW | AdamW | AdamW | AdamW |
| Learning Rate | 1e-5 | 2e-5 | 8e-5 | 2e-5 |
| Layer LR Deacy | 0.9 | – | – | – |
| Weight Decay | 0.01 | 0.01 | 0.01 | 0.01 |
| Training Steps | 50000 | 750,000 | 1,000 | 600,000 |
| Warmup | 5000 | 75,000 | 0 | 10,000 |
| Token Batch Size | 2048 | 2048 | 4096 | 4096 |
| Max Sequence Length | 1024 | 1024 | 1024 | 1024 |
| EL passages | 100 | – | – | 25 |
| RE passages | – | 24 | 5 | 20 |

Table 4: Hyperparameter we used to train the Reader for Entity Linking (AIDA), Relation Extraction (NYT) and cIE (REBEL).

## A.3 EFFICIENCY

Efficiency is a crucial factor in the practical deployment of Information Extraction systems, as real-world applications often require rapid and scalable information extraction capabilities. ReLiK excels in this regard, outperforming previous systems in performance, memory requirements, and speed. Table 5 shows the training and inference speeds of ReLiK.

**EL** Until now, efficiency had been a clear bottleneck for most EL systems, which rendered them useless or highly expensive on real-world applications. Therefore the efficiency gains for EL were extensively discussed in the main body of the paper at Section 4.2.

**RE** On the RE side, the only system on-par in terms of speed and performance would be USM. Unfortunately, it is not openly available, limiting its utility for the broader research community and hindering our ability to asses its speed. In Section A.6 we discuss other of its shortcomings. Instead, Table 5 compares the current openly available RE system with the best performance on NYT, REBEL. As an autoregressive system, inference speeds are several orders of magnitude higher. $ReLiK_L$ outperforms it by more than 2 F1 points and it is still around 3x faster, while $ReLiK_L$, which still outperforms any previous system, takes only 10s, a 10x gain in terms of speed.

**cIE** ReLiK continues to shine in the domain of closed Information Extraction, where it outperforms existing systems in terms of efficiency and performance. Compared with two other leading systems, $ReLiK_S$ surpasses them in F1 score while significantly outpacing them in terms of speed. These systems rely on BART-large, making them several orders of magnitude slower. In Table 5 we

| | Train | | | | | |
|---|---|---|---|---|---|---|
| | Retriever | $\text{ReLiK}_S$ | $\text{ReLiK}_B$ | $\text{ReLiK}_L$ | Previous SotA | GPU |
| AIDA (EL) | 4 h | – | 11h | 36h | 48 h | A100 |
| NYT (RE) | 2 h | 14 h | 21 h | 48 h | 34 h | 3090 |
| REBEL (cIE) | 6 h | 20 h | 30 h | 3 d | 18.5 d | V100 |
| | Inference | | | | | |
| AIDA (EL) | 6 s | – | 23s | 100s | 20 m | A100 |
| NYT (RE) | 2 s | 8 s | 14 s | 28 s | 105 s | 4090 |
| REBEL (cIE) | 5 m | 10 m | 17 m | 36 m | 10 h | 4090 |

Table 5: Training and inference times for ReLiK on a single NVIDIA RTX 4090 GPU. Retriever times are reported separately, as they are shared across Reader sizes. The total time for any model size X is Retriever + $\text{ReLiK}_X$. Results for previous SotA (State-of-the-Art) in the right side are taken from the best performing openly available systems trained on each dataset and task. Zhang et al. (2022, entQA) for AIDA, Huguet Cabot & Navigli (2021, REBEL) for NYT and Josifoski et al. (2022, GenIE) for REBEL. Inference times refer to the time needed to annotate the corresponding test split for each dataset.

| Model Name | Recall@100 | Recall@50 |
|---|---|---|
| Baseline | 81.9 | 71.6 |
| + Hard-Negatives | 98.5 | 97.9 |
| + Document-level information | 98.8 | 98.0 |
| + BLINK Pretrain | 99.2 | 98.8 |

Table 6: Ablation for the Retriever module. Each line represents an additional change built upon the previous one.

report on GenIE as its inference and train time are reported, but it should be noted that both GenIE and KnowGL are roughly equivalent in terms of compute. Here, again, the speed gains are multiple orders of magnitude, from 40x with $\text{ReLiK}_S$ to 15x with $\text{ReLiK}_L$.

In conclusion, ReLiK redefines the efficiency landscape in Information Extraction. Its unified framework, reduced computational requirements, and speed make it a compelling choice for a wide range of IE applications. Whether used in research or practical applications, ReLiK empowers users to extract valuable information swiftly and efficiently from textual data, setting a new standard for IE system efficiency.

| | EL | | | RE | | | | |
|---|---|---|---|---|---|---|---|---|
| K | 100 | 50 | 20 | 24 | 16 | 12 | 8 | 4 |
| $\text{ReLiK}_S$ | — | — | — | 94.4 | 94.5 | 94.5 | 94.5 | 94.2 |
| Time | — | — | — | 10 s | 10 s | 10 s | 8 s | 6 s |
| $\text{ReLiK}_B$ | 85.9 | **86.1** | 85.8 | 94.8 | 94.8 | 94.8 | 94.8 | 94.5 |
| Time | 23 s | 14 s | 6 s | 14 s | 14 s | 12 s | 10 s | 9 s |
| $\text{ReLiK}_L$ | **86.5** | 86.5 | 86.1 | 95.1 | **95.2** | 95.1 | 95.1 | 94.8 |
| Time | 100 s | 47 s | 22 s | 28 s | 24 s | 22 s | 20 s | 18 s |

Table 7: Micro-F1 results and inference time on AIDA for EL and NYT for RE when we reduce the number of retrieved passages as input to the Reader. Times reported are just for the Reader, without the retrieval step. Notice that for $K = 24$, all relation types in NYT are part of the input.

## A.4 Ablations

### A.4.1 Entity Linking

**Retriever** Table 6 presents the findings of our ablation study conducted on the Retriever using the validation set from AIDA. In the baseline configuration, we initialize the model with $E5_{base}$ and train it by optimizing the loss (1) with a focus solely on in-batch negatives. The introduction of hard-negatives substantially improve recall rates. Additionally, document-level information proves beneficial to the Retriever, albeit particularly benefiting AIDA, where relevant information is concentrated in the first token. Furthermore, the pretraining on BLINK demonstrated significant impact, especially on Recall@50, suggesting that pretraining enhances the Retriever ability to efficiently rank the candidate entities.

**Passages Trimming** The Retriever serves as a way to limit the number of passages that we consider as input to the Reader. At train time, we set $K = 100$, which, as Table 6 just showed, has a high Recall@K. However, as the computational cost of the Transformer Encoder that serves as the Reader grows quadratically on the input length, the choice of $K$ affects efficiency. Table 7 shows what happens when we reduce the number of passages at inference time. Surprisingly, performance is not affected; in some cases, it even improves while time is halved. This showcases the usefulness of the Retriever which while fast is still able to rank passages effectively.

### A.4.2 Relation Extraction

**No Retriever** Our benchmarks for RE contain a small number of relation types (5 and 24). Therefore the Retriever component is not extrictly necessary when all types fit as part of the input. Still, we believe it is an important part of the RE pipeline, as it is more flexible and robust to cases outside of the benchmarks. For instance, in long-text RE where the input text is longer, there is a need to reduce the number of passages as input to the Reader. Or as the case with cIE with REBEL, when the relation type set is larger, the Retriever enables an unrestricted amount of relation types. Nevertheless, we assess the influence of the Retriever as a reranker for NYT and explore a version of ReLiK without a Retriever. To do so we train a version of our Reader where the relation types are shuffled (ie. without a Retriever step). We obtained a micro-F1 of 94.2 for $ReLiK_S$, which is just slightly worse. Given how fast the Retriever component is at inference time, this result showcases how even when not strictly needed, it does not hurt performance.

**Passages Trimming** The previous section seemed to indicate that for datasets with a small set of relation types there is no need of a Retrieval step and a standalone Reader would be enough. While this is certanly an option, the Retrieve step is still very fast and doesn't add much overhead computation. On the other hand, the Reader is considerably slower, as the input is larger with additional computation that adds to the overall computational time. For RE the Hadamard product step grows quadratically with the number of passages. Therefore we explore how it affects downstream performance to reduce the number of passages once the system is already trained. We want to find out 1) is performance affected 2) is it considerably faster to reduce the number of passages. As Table 7 shows, reducing the number of passages up to just 8 doesn't impact performance. In fact, we even obtained better results with just 16 passages instead of 24.

**Entity Linking as an aid to Relation Extraction** On the cIE setup where Entity Linking and Relation Extraction are performed by the same Reader, each task is performed sequentially and then RE predictions are conditioned on EL. But does EL aid RE? Or does having a shared Reader between both tasks impact RE negatively? Entity types were often included in Relation Classification to improve the overall performance Zhou & Chen (2022). In our case, RE is conditioned on EL implicitly, without explicit ad-hoc information, i.e. just by leveraging the predictions of the EL component. We train $ReLiK_S$ on REBEL without EL, which performs solely RE under the same conditions and hyperparameters as the cIE counterpart. The system without EL obtained a micro-F1 of 75.4 with boundaries evaluation. On the other hand, the cIE approach that combines both EL and RE, we obtain 76.0 micro-F1[6], which considering the size of the test set (175K sentences) is

---

[6]This value differs from the one reported in Table 2 since it is evaluated without entity disambiguation

| System using BERT-base | P | R | F |
|---|---|---|---|
| (Sui et al., 2023) | 92.5 | 92.2 | 92.3 |
| (Zheng et al., 2021) | 93.5 | 91.9 | 92.7 |
| (Lou et al., 2023, USM$_{BERT-base}$) | **93.7** | 91.9 | 92.8 |
| ReLiK$_{BERT-base}$ | 93.2 | **92.9** | **93.1** |

Table 8: Results for systems using BERT-base on the NYT dataset.

a considerable difference. This is an exciting result as it validates end-to-end approaches for cIE where both tasks are combined.

**BERT-base**   Our Reader is based on DeBERTa-v3, while previous RE systems may be based on older models. To enable a fair comparison and assess the flexibility of our RR approach, we train our Reader on NYT using BERT-base and compare with other systems. Table 8 shows how ReLiK$_{BERT-base}$ outperforms previous approaches, including USM.

## A.5   ERROR ANALYSIS

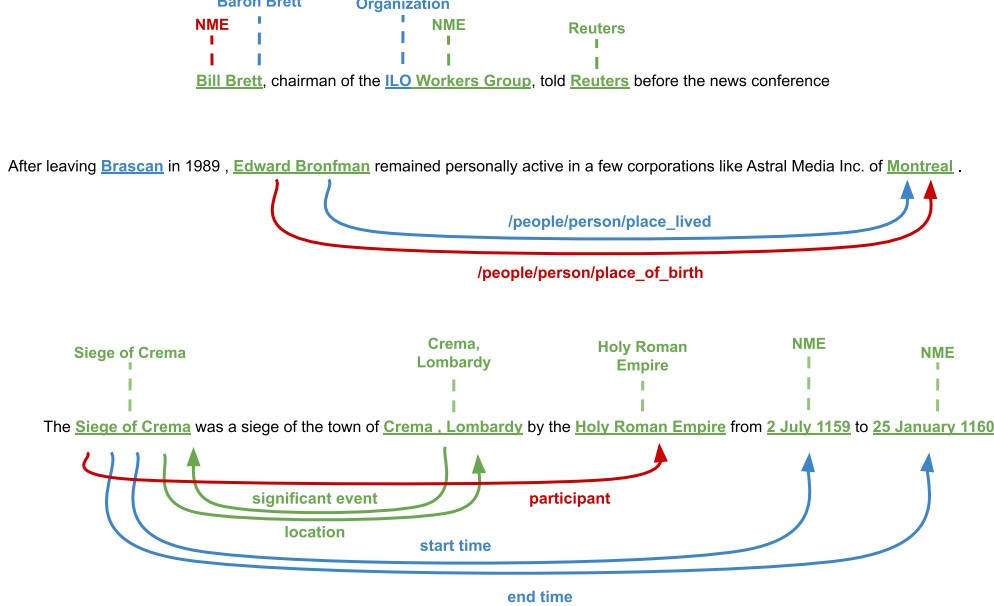

Figure 2: Example predictions by ReLiK$_L$ on AIDA (top), NYT (middle), and REBEL (bottom) for EL, RE, and cIE respectively. Green stands for true positive, blue for false positive, and red for false negative.

**Entity Linking**   Figure 2 shows an example of the predictions generated by our system when trained on EL. This particular example showcases a common error when evaluating the AIDA dataset. AIDA was manually annotated in 2011 on top of a Named Entity Recognition 2003 dataset (Tjong Kim Sang & De Meulder, 2003). While widely used as the de-facto EL dataset, it contains errors and inconsistencies. A common one is the original entity spans not being linked to any entity in the KB. This could either be because at the time such an entity was not present in the KB, or an annotation error due to the complexity of the task. This leads to NME annotations which at evaluation time are considered false positives, as our system links to the correct entity, such as *Bill Brett* in the example. Another source of errors is document slicing in windows. While necessary to overcome the length constraints of our Encoder, it can lead to inconsistent or incomplete predictions. For

instance, *ILO* was linked to an entity in a window that did not see further context (*Workers Group*), while the next window correctly identified *ILO Workers Group* as an NME.

**Relation Extraction**    The example shown in Figure 2 is a common error found in predictions on NYT by ReLiK. Due to the semiautomatic nature of NYT annotations, some relations, such as the ones shown in the example, lack the proper context to ensure consistency at inference time. In this case, the system predicts a relation (*place_lived*) which cannot really be inferred from the text or is ambiguous at best. We believe this is due to certain biases introduced at training time. This can be exemplified by the false negative, annotated as correct (*place_of_birth*), which is impossible to infer from the sentence.

**closed Information Extraction**    Finally, the last example in Figure 2 shows a prediction by our model when trained on both tasks simultaneously with the REBEL dataset. Notice the missing prediction (*participant*), and the false positives. While the passages retrieved contained all the necessary relation types, the system still failed to recover one of the gold triplets, even if all the spans were correctly identified. Then, for the two false positives, while they were not annotated in the dataset, probably due to its automatic annotation, they are correct, and ReLiK predicted them even if, at evaluation time, this will decrease the reported performances.

## A.6   USM

In this section, we want to discuss in detail how ReLiK compares with USM. USM is the current state-of-the-art for RE and was the first modern RE system that jointly encoded the input text with the relation types, breaking from ad-hoc classifiers with weak transfer capabilities or autoregressive approaches that leverage its large language head but are inefficient. Therefore, it shares a similar strategy to our RE component, in that both rely on the relation types being part of the input, and the core idea is to link mention spans to their corresponding triplet. However, this is where the similarities end. In USM, the probabilities of a mention span being linked to a triplet (i.e. to another entity and a relation type) are assumed to be independent and factorized such that they are computed separately, in a pairwise fashion. Mentions are linked as subjects to the spans that share a triplet (blue lines in Figure 3) and to the relation type label (green lines). Finally, labels are linked to the object entity (red lines). In most cases, these are sufficient to decode each triplet but we want to point out a shortcoming of this strategy. The decoding is done by pairs. First mention-mention, i.e. in Figure 3 (Jack, Malaga), (Jack, New York), (John, Malaga) and (John, New York); then label-mention (birth place, Malaga), (birth place, New York), (live in, Malaga) and (live in, New York); and finally mention-label (Jack, birth place), (Jack, live in), (John, birth place), (John, live in). At this point, the issue should be clear. From this set of pairs, one cannot retrieve the correct triplets, even though the model would have not made any mistake in its predictions. It is worth pointing out that these phenomena do not happen on either test set for NYT or CONLL04, therefore it doesn't affect reported performance.

John was born in New York, and lives in Malaga while Jack, the other way around. birth place | live in

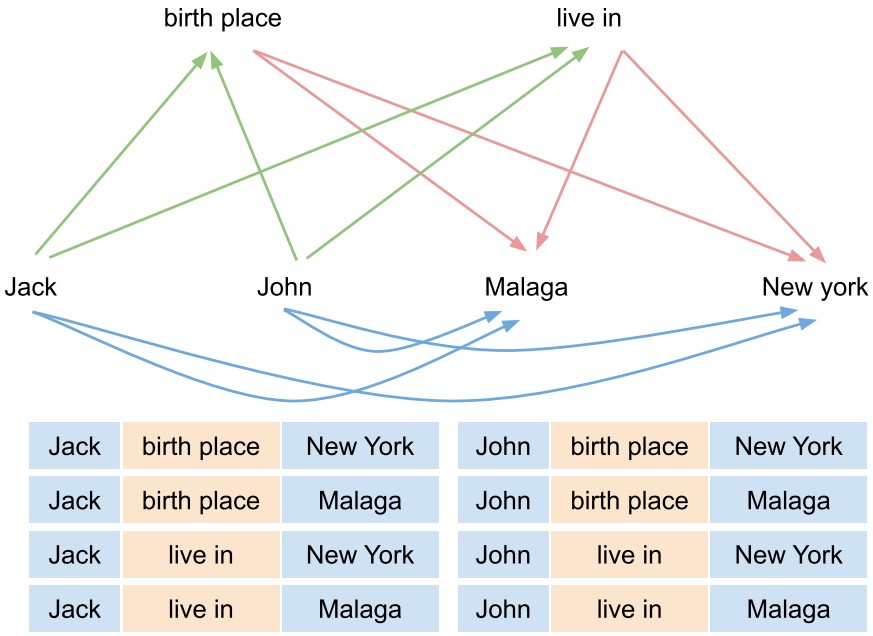

Figure 3: Example of a sentence as input to USM where their token-linking strategy would fail even if the model made the right predictions.

