# OpenReview forum: "ReLiK: Retrieve, Read and LinK: Fast and Accurate Entity Linking and Relation Extraction on an Academic Budget"
_ICLR.cc/2024/Conference — Submitted to ICLR 2024_

### Official Review · Reviewer_5PKz · 2023-10-30

**Soundness:** 2 fair
**Presentation:** 3 good
**Contribution:** 3 good
**Rating:** 6
**Confidence:** 4

**Summary:**

The paper proposes ReLiK, a retriever-reader model for entity linking and relation extraction. ReLiK encodes input text with retrieved candidate entities/relations and can link entities or extract relations in one pass. ReLiK achieves state-of-the-art results on multiple benchmarks while being faster, more parameter efficient, and trainable on a smaller budget than prior art.

**Strengths:**

1. The paper generally has a good presentation that clearly allows readers to understand what was done.
2. ReLiK establishes state-of-the-art results on benchmarks for entity linking and relation extraction. The joint model for closed IE is also insightful.
3. ReLiK is faster than prior state-of-the-art models, with gains of 10-40x reported on inference speed. This makes it much more usable in real applications.

**Weaknesses:**

1. Although ReLiK integrates entity linking and relation extraction together into one framework, the design for each module is relatively simple and similar to previous works.
2. As one of the emphases of this paper is the integration of EL and RE tasks. The mutual influence between EL and RE should be more clearly demonstrated in the experimental analysis section.

**Questions:**

1."Recent approaches only focus on at most two out of the three properties simultaneously." I don't quite understand this sentence. What are the "three properties" referring to？
2. I recognize the efficiency gains achieved by linking entities and extracting relations in just a single forward pass. Yet, I'm curious about what is the core design that enables the model to achieve state-of-the-art performance?I recognize the efficiency gains achieved by linking entities and extracting relations in just a single forward pass. Yet, I'm curious about what is the core design that enables the model to achieve state-of-the-art performance?

---

> ### Author Response · Authors · 2023-11-14
> **Response to Reviewer 5PKz**
>
> We first want to thank you for your review. We hope to address some of the concerns and answer your questions.
>
> **Weaknesses**
>
> > “Although ReLiK integrates entity linking and relation extraction together into one framework, the design for each module is relatively simple and similar to previous works.”
>
> To the best of your knowledge, we are the first to design an architecture capable of performing EL and RE in a single forward step in the Retriever-Reader paradigm, while achieving state-of-the-art performances and inference speed. Furthermore, **simplicity, in our view, is not a drawback**. While individual components might bear resemblance to existing models (such as using a Retriever for EL), the real innovation of ReLiK lies in the synergy of these components, which, to the best of our knowledge, we are the first to consider for closed Information Extraction. Moreover, the modularity of ReLiK is a key advantage with respect to other approaches, since ReLiK can be easily enhanced and expanded by leveraging any improvement on each of its components separately.
>
> > “As one of the emphases of this paper is the integration of EL and RE tasks. The mutual influence between EL and RE should be more clearly demonstrated in the experimental analysis section.”
>
> We agree, and that is why **we explored such interaction in Appendix A.4.2** and showed how EL helped RE, evidenced by a notable drop in F1 score when RE is isolated. Performance in Table 2 for EL is reported solely on entities present in triplets, but if we check the F1 score for entities in general, validation F1 reached 87.46 for ReLiK small. We ran an experiment where EL was trained with the same data without RE (similar to the experiment reported in A.4.2) and F1 reached 87.5. While, in this case, RE doesn’t seem to provide a benefit to EL, it does not affect negatively either. We would have liked to devote more space to such interactions in the main body; however, in its current shape, the paper already contains a vast array of experiments and ablations that we considered more relevant, at least within the main body. We welcome suggestions for additional experiments to explore this interaction further, and we will happily provide results if there’s time during this rebuttal period, or in the final version of the paper otherwise.
>
> **Questions:**
> > "Recent approaches only focus on at most two out of the three properties simultaneously." I don't quite understand this sentence. What are the "three properties" referring to？
>
> It refers to “Speed, Flexibility, and Performance.”, which are mentioned in the previous paragraph. We will make it more clear.
>
> > “I recognize the efficiency gains achieved by linking entities and extracting relations in just a single forward pass. Yet, I'm curious about what is the core design that enables the model to achieve state-of-the-art performance?”
>
> **The inclusion of verbalized labels in the input was crucial**, enabling not only the efficiency of a single forward pass but also enhancing overall performance. We theorize that the Reader benefits from contextualizing the linking task with all candidates, rather than considering each in isolation, as some previous work did. This approach allows the Reader to leverage both the other candidates as well as the order in which they are given by the Retriever. Our ablation studies in A.4.2 show a performance decline when the Retriever's order is disregarded for the RE task. This drop was even more significant for EL, 85.8 for ReLiK Large, vs. 86.5 F1 reported in Table 2, a 0.7 drop which we will report in the final version of the paper. This indicates the Retriever's dual role in both filtering and ranking, which significantly contributes to ReLiK's effectiveness. Moreover, it is also important to point out that using a verbalized label as part of the input allows the model to be more flexible for unseen entities, as the hidden representations are always contextual, whether seen during training or not, rather than fixed for each entity. We believe this is one of the key reasons of the strong Out-of-domain performance reported in Table 2.

---

### Official Review · Reviewer_2r4r · 2023-11-01

**Soundness:** 2 fair
**Presentation:** 3 good
**Contribution:** 3 good
**Rating:** 6
**Confidence:** 3

**Summary:**

This article introduces Retriever & Reader pipeline to Entity Linking (EL) and Relationship Extraction (RE) tasks. ReLiK uses retriever instead of classifiers to discover entities and entity relationships in text, the reader module's role is to identify relevant entities or relations retrieved and align them with the corresponding textual spans. The experimental results show that the proposed methodology strikes a balance between effectiveness and efficiency.

**Strengths:**

The proposed method has several practical advantages. The reader greatly improve the efficiency of entity linking and relationship extraction.

Empirical evaluation thoroughly covers a substantial number of datasets.

**Weaknesses:**

The baseline systems used for comparison were not comprehensive enough in Section 4,  and it was recommended that more baseline systems be added for comparison[1], [2], [3].

Since the contribution of this paper lies in the novel paradigm, the authors could have devoted a chapter to a brief overview of the developmental lineage of the relevant paradigm in order to describe more clearly the special features of this paper.

[1] Johannes M. van Hulst, Faegheh Hasibi, Koen Dercksen, Krisztian Balog, and Arjen P. de Vries. 2020. REL: An Entity Linker Standing on the Shoulders of Giants. In *Proceedings of SIGIR*

[2] Nikolaos Kolitsas, Octavian-Eugen Ganea, and Thomas Hofmann. End-to-end neural entity linking. In *Proceedings of the 22nd Conference on Computational Natural Language Learning*

[3] Johannes Hoffart, Mohamed Amir Yosef, Ilaria Bordino, Hagen Fu ̈rstenau, Manfred Pinkal, Marc Spaniol, Bilyana Taneva, Stefan Thater, and Gerhard Weikum. Robust disambiguation of named entities in text. In *Proceedings of the EMNLP*

**Questions:**

The author mentioned that “ReLiK excels in this regard, surpassing previous systems in terms of performance, memory requirements, and speed”, but did not provide a quantitative comparison of memory requirements with other systems.

In the textual description of Section 3, the word "passage" is confusing - does it refer to the entities and relationships obtained by the retriever? The authors need further clarification.

Is there error propagation when the retriever fails to retrieve relationships and entities from top-k results?

---

> ### Author Response · Authors · 2023-11-14
> **Response to Reviewer 2r4r**
>
> Thank you for the time taken to review our work. We hope we can address your concerns.
>
> **Weaknesses**
>
> > “The baseline systems used for comparison were not comprehensive enough in Section 4, and it was recommended that more baseline systems be added for comparison[1], [2], [3].”
>
> Unfortunately, we prioritized including the most current and impactful EL systems for our baseline/comparison systems due to the limited space available. The systems suggested by the reviewer, although valuable, exhibit lower performance than our baselines. However, we will include these lower baselines in a more comprehensive EL comparison in the Appendix.
>
> > “Since the contribution of this paper lies in the novel paradigm, the authors could have devoted a chapter to a brief overview of the developmental lineage of the relevant paradigm in order to describe more clearly the special features of this paper.”
>
> Again, space constraints led us to focus on the previous trends and systems for each task in sections 4.1.2  and 5.1.2, but we did not have the space to delve into the explanation of Retriever-Reader systems in general. We will add a more detailed explanation of the “developmental lineage” of RR systems in the Appendix.
>
> **Questions:**
> > “The author mentioned that “ReLiK excels in this regard, surpassing previous systems in terms of performance, memory requirements, and speed”, but did not provide a quantitative comparison of memory requirements with other systems.”
>
> We reported on the GPU requirements of each system, which is indirectly related to memory. However, we will make sure to include a more quantitative comparison for memory. For our systems, training never requires more than 24GB of memory. At inference time, the retriever with e5 base and batch size of 128 uses 881 MB of VRAM. The reader on its larger model (debate-v3-large) with a batch size of 4096 tokens uses 3985 MB of VRAM for EL, 4497 MB of VRAM for RE. As for the passages index for the retriever, it takes 10 GB of VRAM if loaded on the GPU with fp16, and 17 GB of RAM if loaded on the CPU with fp32. For comparison, [Zhang et al. (2022)] index takes 22 GB of RAM with fp32.
>
> > “In the textual description of Section 3, the word "passage" is confusing - does it refer to the entities and relationships obtained by the retriever? The authors need further clarification.”
>
> **Short answer: Yes.** More in general, a “query” is the input text that will be annotated, while the “passages'' are the textual representations of the entities or relations to be retrieved using the input query. In Section 3, we adopted the terminology from DPR [Karpukhin et al., 2020] to maintain consistency with prior literature on retriever systems and, therefore, not to be specific on the tasks tackled later, to avoid referring to entities and relations and keep the description of the Retriever-Reader paradigm as general as possible. However, we do understand that it might be confusing. Thanks for pointing it out. We will clarify this in the final version of the paper.
>
> > “Is there error propagation when the retriever fails to retrieve relationships and entities from top-k results?”
>
> Yes, whenever the Reader component fails to retrieve an entity/relation, the Reader component cannot link/extract it, and thus either it does not predict it at all (false negative) or it links/extracts a wrong one. However, the retriever has a very high recall, as shown in Table 6 of the Appendix: 99.2 R@100 and 98.8 R@50, and it rarely is the source of errors. We didn’t report on recall for cIE in the paper, but it is R@20 of 99% for relations and R@25 of 98% for entities on the REBEL dataset. We will add these numbers to the paper as well.

---

### Official Review · Reviewer_5D75 · 2023-11-02

**Soundness:** 2 fair
**Presentation:** 3 good
**Contribution:** 2 fair
**Rating:** 5
**Confidence:** 4

**Summary:**

This paper proposes ReLiK, a new Retriever-Reader architecture for EL and/or RE. Given an input text, ReLiK allows to extract relations between entities given a reference knowledge base in a single forward pass. The proposed approach achieves state-of-the-art performance for the closed information extraction task (EL + RE) on standard datasets.

**Strengths:**

The proposed approach offers fast inference and state-of-the-art performance at a reasonably low budget, which is important for various settings. The paper is well-written and easy to follow. The adaptation of the Retriever-Reader paradigm to cIE is original and, to the the best of my knowledge, has not been proposed before.

**Weaknesses:**

The proposed approach is underpinned by access to external knowledge since ReLiK is given as input the text together with entities and relations from the KB. This impacts the performance and efficiency of the model and raises concerns about the fairness of the proposed benchmarks.

More specifically, the fact that ReLiK relies on the entities and relations from the KB already provides the model with the set of possible entities that can be extracted from the text, which can help for demarcating and disambiguating entities, and also for extracting relations.

The access to this non-parametric memory is also what enables to considerably lower the number of parameters, thereby offering faster inference time.

Also, the following recent prior work [1], which uses an end-to-end Reader-Retrieval approach for EL, should be cited in the paper. It would be interesting to see how both methods compare.


Minor comments:

- Xs and Xt are not defined in section 3.2, in the definition.
- The wrong template was used for submission (ICLR 2023)

[1] Bidirectional End-to-End Learning of Retriever-Reader Paradigm for Entity Linking, Li et al., arXiv:2306.12245, 2023

**Questions:**

What is the point of $<ST_{0}>$ ? It is not associated with any passage and we already have the [SEP] special token to dissociate between the text and the retrieved passages.

---

> ### Author Response · Authors · 2023-11-14
> **Response to Reviewer 5D75**
>
> We first want to thank you for your review and hope to answer your questions as well as clear some possible misunderstandings.
>
> **Weaknesses**
>
> The statements:
>
> >  1. “The proposed approach is underpinned by access to external knowledge since ReLiK is given as input the text together with entities and relations from the KB. **This impacts the performance and efficiency of the model and raises concerns about the fairness of the proposed benchmarks.**”, and
> 2. “More specifically, the fact that ReLiK relies on the entities and relations from the KB already provides the model with the set of possible entities that can be extracted from the text, which can help for demarcating and disambiguating entities”
>
> Makes us believe this viewpoint might stem from a misunderstanding, which we are eager to resolve. **Every system under comparison uses external knowledge and has access to the possible entities that can be extracted from the text**. Specifically, systems in [De Cao et al. 2021 a] and [De Cao et al. 2021b] both use an external mention-entity index in which, as stated in our paper, every possible mention (e.g., Barack Obama or Washington) is associated with the list of possible entity classes (i.e. Wikipedia titles) that can be associated with them. [Zhang et al. (2022)] and [1], sharing our Retriever approach, access the same non-parametric external knowledge as ours and the set of extractable entities from the text. Therefore, within the tasks of EL and RE, **external knowledge in the form of entity and relation titles, or their definitions, is not a source of unfairness but rather available information for the task**, analogous to a training set with labels. Finally, along with the previous reasoning, we want to be very clear that, **at inference time, ReLiK receives the text as input, and solely the text**. It is the combination of the Retriever and the Reader that is able first to reduce the number of candidates and then link them. The initial search space encompasses the entire set of entities or relations, as is standard in EL and RE systems, thus ensuring fairness in task comparisons against other systems.
>
>
> > *“Also, the following recent prior work [1], which uses an end-to-end Reader-Retrieval approach for EL, should be cited in the paper. It would be interesting to see how both methods compare.”*
>
> [1] was published in July 2023 as a preprint on Arxiv, with the deadline for ICLR 2024 being in September, and therefore, we were not able to discuss it in the paper at submission time. It is an extension of [Zhang et al. (2022)] with an interesting new end-to-end training approach. Unfortunately, they do not report results on datasets other than AIDA, and the absence of publicly available code impedes direct comparison. Their results on AIDA are in the same ballpark as ours, but the model suffers from the same shortcomings as [Zhang et al. (2022)] in terms of efficiency, or even worse, as they discuss in their Limitations section. For instance, their system requires 50GB of GPU memory, and takes 3.5 hours per epoch on 4 A100 GPUs, while ours can be trained on a single 4090 and takes half an hour per epoch for the Retriever, and the same for the Reader, on its largest version. Exploring the potential integration of our lightweight paradigm into their end-to-end approach remains an intriguing prospect. We will make sure to cover this paper in the final version of the paper.
>
> **Minor Comments**
> > Xs and Xt are not defined in section 3.2, in the definition.
>
> We will make it more explicit that *s* and *t* denote the indices of start and end tokens of a span in *X*.
>
> > The wrong template was used for submission (ICLR 2023)
> We are sorry for using the wrong year, and thanks for noticing it. We will update the year number. Luckily, except for the year, the template itself is the same as the 2024 one.
>
> **Questions**
>
> > What is the point of ⟨ST$_0$⟩? It is not associated with any passage and we already have the [SEP] special token to dissociate between the text and the retrieved passages.
>
> This is explained in the third footnote, found in page 4:
> Here e$_0$ symbolizes NME, i.e. mentions for which the gold entity is not in E, represented by ⟨ST$_0$⟩
>
> Therefore, it is used for EL and cIE when a span cannot be linked to any entity candidate because either it is not in the KB or the Retriever module failed to retrieve it, and therefore it is not present in the candidates. To avoid confusion we will make sure to mention it earlier, where it is first shown in Equation 2.

---

> > ### Author Response · Authors · 2023-11-14
> > **Papers from Response to Reviewer 5D75**
> >
> > **Cited Papers**
> >
> > > Nicola De Cao, Wilker Aziz, and Ivan Titov. Highly parallel autoregressive entity linking with discriminative correction. In Marie-Francine Moens, Xuanjing Huang, Lucia Specia, and Scott Wentau Yih (eds.), Proceedings of the 2021 Conference on Empirical Methods in Natural Language Processing, EMNLP 2021, Virtual Event / Punta Cana, Dominican Republic, 7-11 November, 2021, pp. 7662–7669. Association for Computational Linguistics, 2021a. doi: 10.18653/v1/2021. emnlp-main.604. URL https://doi.org/10.18653/v1/2021.emnlp-main.604.
> >
> > > Nicola De Cao, Gautier Izacard, Sebastian Riedel, and Fabio Petroni. Autoregressive entity retrieval.
> > In 9th International Conference on Learning Representations, ICLR 2021, Virtual Event, Austria, May 3-7, 2021. OpenReview.net, 2021b. URL https://openreview.net/forum?id=5k8F6UU39V
> >
> > > Wenzheng Zhang, Wenyue Hua, and Karl Stratos. EntQA: Entity linking as question answering. In International Conference on Learning Representations, 2022. URL https://openreview.net/forum?id=US2rTP5nm_.

---

### Author Response · Authors · 2023-11-18
**Request for engagement**

We wish to highlight to the Reviewers that we have thoroughly addressed their inquiries. We kindly seek further engagement to clarify specific misunderstandings and to receive acknowledgment regarding the effectiveness of our responses in alleviating their concerns. Specifically, we have provided detailed explanations to address the fairness concerns raised by Reviewer 5D75, stemming from a misunderstanding. Additionally, we have highlighted that the baseline systems suggested by Reviewer 2r4r underperform in comparison to the systems we have already reported on. We believe that engaging in a proactive dialogue on these and other key points will not only contribute to the enhancement of our work but also facilitate a more comprehensive evaluation on their part.

---

### Meta-Review · Area_Chair_2WuG · 2023-12-04

**Metareview:**

The paper introduces ReLiK, a retriever-reader model for joint entity linking and relation extraction. Using retrieved knowledge, the input is first enriched,  with the retrieved candidate entities and relations, enabling it to link entities and extract relations in a single pass.
The use of the retriever-reader approach for this task is interesting.

However, while ReLiK's integration of entity linking and relation extraction is commendable, the architecture of each module within it remains maybe too similar to existing models. Furthermore, the aspect of doing joint IE is also mature, and not novel.

**Justification For Why Not Higher Score:**

While ReLiK's integration of entity linking and relation extraction is commendable, the architecture of each module within it remains maybe too similar to existing models. Furthermore, the aspect of doing joint IE is also mature and not novel.

**Justification For Why Not Lower Score:**

N/A

---

### Decision · Program_Chairs · 2024-01-16

Reject